# Early-Life Iron Deficiency Anemia Programs the Hippocampal Epigenomic Landscape

**DOI:** 10.3390/nu13113857

**Published:** 2021-10-28

**Authors:** Amanda K. Barks, Shirelle X. Liu, Michael K. Georgieff, Timothy C. Hallstrom, Phu V. Tran

**Affiliations:** Department of Pediatrics, University of Minnesota, Minneapolis, MN 55455, USA; barks012@umn.edu (A.K.B.); liu00459@umn.edu (S.X.L.); georg001@umn.edu (M.K.G.); halls026@umn.edu (T.C.H.)

**Keywords:** perinatal iron deficiency, neurodevelopment, epigenetics, JARIDs, TETs, cognition

## Abstract

Iron deficiency (ID) anemia is the foremost micronutrient deficiency worldwide, affecting around 40% of pregnant women and young children. ID during the prenatal and early postnatal periods has a pronounced effect on neurodevelopment, resulting in long-term effects such as cognitive impairment and increased risk for neuropsychiatric disorders. Treatment of ID has been complicated as it does not always resolve the long-lasting neurodevelopmental deficits. In animal models, developmental ID results in abnormal hippocampal structure and function associated with dysregulation of genes involved in neurotransmission and synaptic plasticity. Dysregulation of these genes is a likely proximate cause of the life-long deficits that follow developmental ID. However, a direct functional link between iron and gene dysregulation has yet to be elucidated. Iron-dependent epigenetic modifications are one mechanism by which ID could alter gene expression across the lifespan. The jumonji and AT-rich interaction domain-containing (JARID) protein and the Ten-Eleven Translocation (TET) proteins are two families of iron-dependent epigenetic modifiers that play critical roles during neural development by establishing proper gene regulation during critical periods of brain development. Therefore, JARIDs and TETs can contribute to the iron-mediated epigenetic mechanisms by which early-life ID directly causes stable changes in gene regulation across the life span.

## 1. Introduction

Increasing evidence indicates that early-life adverse events (e.g., malnutrition) can have a lasting impact on physiological and mental health [1,2]. One particularly well-studied early-life adverse event is iron deficiency (ID), which affects 40–50% of pregnant women and preschool-aged children and is the most common micronutrient deficiency worldwide [3,4]. Given the prevalence of this early-life nutritional exposure, it is important to understand its effect on long-term health outcomes, and the biological basis underlying these effects. ID during the fetal and early childhood periods has a significant effect on neurodevelopment, resulting in cognitive, socio-emotional, and learning and memory deficits that last into early adulthood [5,6]. ID also carries long-term health risks, including increased risk for neuropsychiatric disorders such as autism and schizophrenia [7,8]. Parallel studies in pre-clinical models have shown that early-life ID results in abnormal hippocampal structure, function, and gene expression acutely during the period of rapid neurodevelopment and persistently into adulthood despite prompt iron therapy after diagnosis [9,10,11,12,13,14,15,16,17,18]. The persistent gene dysregulation likely drives the adult neurobehavioral abnormalities of developmental ID. However, specific iron-dependent mechanisms by which early-life ID alters gene expression across the lifespan are unknown. Thus, there is a need to understand the mechanisms by which early-life ID alters gene regulation in the developing brain and leads to permanent changes in the adult brain in order to design better therapeutic strategies to prevent and treat them.

Epigenetic modification is one mechanism by which environmental insults such as ID can enact permanent changes in gene expression [19,20,21,22]. Our group has previously demonstrated that one major epigenetic mechanism, histone methylation, is profoundly altered by ID via modulation of iron-containing and dependent jumonji and AT-rich interaction domain-containing (JARID) histone demethylases [23,24]. In addition, we have shown that early-life ID alters DNA methylation in the hippocampus, although the iron-dependent mechanism of these changes is unknown [25]. In this regard, the iron-dependent Ten-Eleven Translocation (TET) proteins [26,27,28], which oxidize methylcytosine to hydroxymethylcytosine and subsequent derivatives, could significantly contribute to the ID-induced DNA methylation changes. However, the effects of ID on the activity of TET proteins and their enzymatic products in the developing brain remain undetermined.

From a treatment standpoint, choline supplementation during late fetal or early postnatal life has been shown to mitigate behavioral abnormalities in models of genetic and early-life environmental insults, including ID [24,29,30,31,32,33]. Like supplements such as folic acid, betaine, vitamin B12, and L-methionine, choline is a methyl donor for S-adenosylmethionine (SAM), which is a substrate for DNA and histone methylation. Accordingly, maternal diets enriched with methyl-donors increase the methylation of DNA and histones in the epigenome of offspring [34,35,36]. In a rodent model of early-life ID, prenatal choline supplementation during the period of high iron demand in hippocampal development rescues recognition memory deficits and *Bdnf* dysregulation in formerly ID anemic rats by reversing underlying epigenetic dysregulation [24,29]. Thus, choline supplementation may be an effective treatment for developmental ID amenia by targeting the epigenetic mechanisms underlying the long-term sequelae of developmental ID anemia.

The persistence of neurocognitive deficits and gene dysregulation despite prompt treatment of ID with iron replacement indicates that better targeting of iron therapy and additional adjunct therapeutic options are needed. Until we have a better understanding of the cellular and molecular mechanisms underlying the lasting functional and gene expression changes associated with early-life ID, we cannot move forward with advancement of more effective prevention and treatment for this common developmental condition because iron repletion alone is not fully effective. The present review summarizes the state of knowledge and identifies gaps in our understanding of the iron-dependent molecular mechanisms underlying the long-term neurological effects of early-life ID.

## 2. Main Text

Iron deficiency during neural development causes long-term neurobehavioral outcomes.

ID is the foremost micronutrient deficiency worldwide, affecting an estimated 40–50% of pregnant women and preschool-aged children [3,37]. While prevalent in low- and middle-income countries, ID is also common in high-income countries, including the US, where up to 42% of pregnant women and 14% of 1–2 year-olds experience ID [38,39,40]. With respect to health disparity, ID affects about 6% of women of child-bearing age in the US. However, among those women affected, Black and Hispanic are twice as likely to be affected compared to White or Asian women [41]. The prevalence of ID among these vulnerable populations is a major public health concern due to the well-documented effects of developmental ID on neurodevelopment.

Individuals who were iron-deficient during infancy exhibit long-lasting neurocognitive and neuropsychiatric abnormalities. Formerly iron-deficient (FID) children show evidence of neurocognitive impairments, including slower perceptual speed, poorer understanding of quantitative concepts, and impaired language abilities [42,43]. FID adolescents and young adults exhibit lower performance in reading and arithmetic tasks [44] as well as recognition memory and strategy shifting [6]. FID adults are prone to not complete secondary education or actively seek additional training [5]. In terms of neuropsychiatric outcomes, FID children exhibit lower positive affect, poorer performance on a delayed gratification task, and more passive or unengaged compared to iron-sufficient children [45,46,47]. Early adolescence FID individuals show an increased propensity for anxiety, depression, and aggression [44]. FID adults are more prone to report negative emotions and poor emotional health [5].

Low maternal iron intake and ID during pregnancy can increase the risk of autism spectrum disorder (ASD) [8] and schizophrenia spectrum disorder in offspring [7,48]. These associations are corroborated by a large-scale cohort study of more than 500,000 individuals where the prevalence of ASD, attention deficit and hyperactivity disorder (ADHD), and intellectual disability are increased in children of mothers diagnosed with anemia during the first 30-weeks of pregnancy [49]. It is noteworthy that the prevalence of these disorders did not increase when the diagnosis of anemia in mothers occurred after 30 weeks of pregnancy, suggesting that neurodevelopmental events during the first two trimesters of pregnancy are most critical for the atypical neuropsychiatric development and highly sensitive to iron insufficiency. Collectively, these clinical studies underscore a role for iron as a critical nutrient across neurodevelopment.

### 2.1. Early-Life Iron Deficiency Modifies Gene Regulation and Epigenetic Landscape in the Adult Rat Hippocampus

Understanding the biology behind the association of maternal and early postnatal ID with neurocognitive impairments and psychopathology risks is a prerequisite for developing effective prevention and treatment strategies. Based on a series of observational studies, two hypotheses are formulated to explicate the poor long-term neurodevelopmental outcomes. The structural defects hypothesis posits that developmental ID causes abnormalities ranging from gross structures (e.g., brain and white matter volumes) to fine ultrastructures (e.g., dendrite branching and synaptic spines) that persist despite later iron reconstitution [13,15,50]. The gene dysregulation hypothesis postulates that early-life ID reprograms gene regulation, which in turn contributes to subsequent phenotypic changes [18,51,52,53,54]. These two hypotheses are not mutually exclusive, and their interactions likely drive abnormal structure during neurodevelopment and function throughout the lifetime, accounting for the risk of psychopathology in later life. The structural hypothesis is discussed elsewhere [55]. This review focuses on the gene dysregulation hypothesis and the potential underlying mechanisms.

### 2.2. Early-Life Iron Deficiency Reprograms Gene Regulation

Extensive gene dysregulation has been demonstrated in both rodent and porcine models of fetal-neonatal ID acutely during ID [10,56] and persistently in adulthood following iron repletion [18,54]. The dysregulated genes implicate abnormal neurodevelopment and increased propensity of neuropsychiatric disorders [10,18,54,56]. These widespread and lasting effects implicate global and stable changes in gene regulatory mechanisms such as epigenetic regulation. Epigenetic regulation refers to covalent modifications of DNA and histones to alter gene transcriptional activity and phenotype without changes in the genetic code. Importantly, DNA and histone modifications can be altered depending on environmental exposures such as stress [57,58], toxicants [59,60,61], and nutrients [24,51,52,53,62,63]. Thus, epigenetic regulation can be a mechanism by which ID alters gene regulation during critical windows of the nervous system development, contributing to poor long-term neuropsychiatric outcomes.

### 2.3. DNA Methylation and Hydroxymethylation

DNA can be methylated by the covalent addition of a methyl group to cytosine nucleotides. Methylated DNA is generally associated with gene silencing, particularly when present at promoter regions, by inducing closed-state chromatin [64]. Both rat and porcine models of developmental ID demonstrate significant changes in DNA methylation at a genome-wide level; however, few of the differentially methylated CpG sites map to differentially expressed genes [25,56]. Thus, while clearly demonstrating altered patterns of DNA methylation, these studies do not address the functional significance of these changes.

Despite the evidence of altered global DNA methylation pattern, the biological mechanism by which fetal-neonatal ID directly mediates these changes remains unclear. A potential consideration is that methylated DNA (5-methylcytosine, 5mC) can be oxidized by the iron-dependent family of TET methylcytosine dioxygenases, which absolutely require iron for their enzymatic activity (Figure 1A). If the iron-binding site is mutated, or if iron is depleted, TET proteins do not efficiently convert 5mC to 5-hydroxymethylcytosine (5hmC), resulting in significantly decreased global 5hmC levels [28]. Interestingly, TET expression increases in the central nervous system (CNS) during development, particularly as neurons mature. In the mouse hippocampus (Figure 1B), 5hmC levels increase significantly (2.6 fold) between postnatal day (P)7 and adulthood, predominantly within the exons of genes that become activated between these two timepoints [65]. Globally, 5hmC is enriched in the intragenic regions of genes that are highly transcribed [66]. Taken together, TET enzymes are highly active during brain development when they actively establish 5hmC levels as a stable epigenetic modification that plays a critical role in the regulation of neural-specific gene expression. In the context of fetal-neonatal ID, little is known about the roles of TETs. Therefore, additional studies are needed to determine the effect of a clinically relevant degree of ID on TET activity in the developing brain.

### 2.4. Histone Methylation

Histones can be acetylated, methylated, phosphorylated, and ubiquitinated. These modifications can alter gene transcriptional regulation by stably establishing activation or repression protein complexes. The histone code is complex and is understudied in the context of early-life ID. In the context of developmental ID, lysine demethylation of histones is particularly important because of its mechanistic dependence on iron (Figure 2A). Removal of methyl groups from lysine residues can be catalyzed by the JARID protein family, which absolutely requires iron for their catalytic activity [68,69]. It is noteworthy that the incorporation of iron into the Jumonji C (JmjC) domain is also critical for JARID’s structural stability and function [68]. JARID protein family, which is also known as KDM (Lysine demethylase), has a strong preference for specific lysine targets, thereby producing differential transcriptional regulatory effects (Figure 2B). For instance, JARID1B (KDM5 family member) removes the methyl group from tri- and di-methylated histone H3 lysine 4 (H3K4me3/2), leading to a less transcriptionally active chromatin conformation; whereas, JMJD3 (KDM6 family member) catalyzes tri- and di-methylated histone H3 lysine 27 (H3K27me3/2), resulting in a less repressed chromatin conformation [70,71]. Moreover, JARIDs regulate neuronal growth and differentiation during brain development [72,73,74,75,76,77]. Importantly, JARIDs can regulate *Bdnf* expression, which exhibits a long-term downregulation by early-life ID [17,75,78,79,80,81]. Fetal-neonatal ID produces a long-term downregulation of hippocampal JARIDs [23]. In addition, the adult formerly iron-deficient hippocampus exhibits lower expression of JMJD3 concomitant with a higher enrichment of H3K27me3 at the *Bdnf* promoter; these changes can account for the *Bdnf* downregulation [23,24]. Thus, the general downregulation of JARID subtypes in the iron-deficient and formerly iron-deficient hippocampi likely disrupts a complex balance between active and repressive chromatin structures, leading to permanent gene dysregulation beyond the critical period of brain development.

Collectively, early-life ID can alter histone methylations, contributing to the long-term gene expression changes in adulthood. Given the complexity of epigenetic regulation, additional studies are needed to fully assess the epigenetic modifications, particularly at gene promoters, in order to elucidate mechanisms contributing to the long-term gene dysregulation.

### 2.5. Prenatal Choline Supplementation and Iron Deficiency Interact to Regulate the Rat Hippocampal Epigenomic Landscape

Given the beneficial effects of prenatal choline supplementation in reversing the repression and epigenetic modifications of the hippocampal *Bdnf* gene by early-life ID [24], it is enticing to use choline as a prevention (prenatal period) or treatment (children diagnosed with ID anemia) in clinical studies [83]. However, there are several caveats to the use of such a potential epigenetic modulator in the treatment of ID. First, given multiple epigenetic modifications by ID, the therapeutic efficacy of modulating any single modification is unclear. Our recent genome-wide analysis reveals that prenatal choline supplementation produces a distinct epigenomic effect from early-life ID, where choline modifies specifically histone H3K9me3 landscape, particularly among loci regulating endocytosis, microgliosis, and neurogenesis in the adult rat hippocampus (Liu and Tran, unpublished observation). Second, the efficacy of epigenetic regulators has to be considered in the context of critical periods of neurodevelopment. For example, once the process of dendrite outgrowth is complete, altering gene expression by treatment with an epigenetic modulator cannot reverse the structural damage caused by ID since the critical period is closed. Therefore, appropriate timing of choline use in the context of neurodevelopment is crucial for its efficacy. Since the brain is not developmentally homogeneous, certain brain regions may be affected more or less than others depending on whether they are or are not in a rapid developmental period. Finally, potential adverse effects of inappropriate use of epigenetic modulators such as choline remain possible. While others have shown the beneficial effects of choline supplementation in normal rats in terms of cognitive function [84,85], we found little and even negative effects in our control iron-sufficient rats using a similar prenatal supplementation (gestational day 11 through 18) paradigm [18,29]. In addition, our recent genome-wide analysis of chromatin accessibility (ATAC) and histone H3K9me3 landscape shows that prenatal choline supplementation caused significant long-term changes in chromatin accessibility and H3K9me3 landscape in the adult rat hippocampus (Liu and Tran, unpublished observation). These observations indicate the need to continue to study appropriate choline dosing, timing, and duration of supplementation to avoid potential long-term adverse consequences.

## 3. Conclusions

The emerging concept of the developmental origins of adult health and disease (DOHaD) posits that the health outcomes in adulthood are determined in part by early-life exposures [86]. Fetal-neonatal ID represents a model of DOHaD, where exposure to ID during development is associated with a lifetime risk of neural dysfunctions, including increased risk for intellectual disability, ASD, ADHD, and schizophrenia [7,8,49]. What mechanisms underlie the effects of fetal-neonatal ID on neurodevelopment to enact the DOHaD concept? While both structural deficits and gene expression reprogramming provide attractive hypotheses to account for lasting functional impairments, it is likely that both factors and their interactive effects contribute to the long-term health outcomes (Figure 3). Emerging evidence implicates the dysregulated iron-dependence epigenetic modifiers (JARIDs, TETs) as key regulators of the persistent epigenetic modifications even after the resolution of ID [24,25,56], providing a causal and direct link between ID and gene dysregulation. In addition, ID-altered epigenome can also play an important role in the structural deficit hypothesis. In CNS development, neuronal migration, axonal growth, and synaptogenesis are regulated by the precisely timed gene expression during specific critical periods of development. These dynamic changes in gene expression are orchestrated in part by the precise changes to the epigenome across neurodevelopment [65,66,87]. If the iron-dependent JARIDs and TETs that orchestrate these developmental epigenetic changes are compromised by fetal-neonatal ID, then the dynamics of gene expression will be compromised as well, consequently leading to abnormal structural development.

Given the global prevalence of fetal-neonatal ID, it is important to establish mechanisms underlying the long-term gene changes relevant to cognitive and emotional dysfunctions due to early-life ID to develop sound nutritional policy and practice that minimizes the risk of long-term neurobehavioral deficits. Prevention of ID during fetal and early postnatal life is an ideal and achievable goal with the provision of optimal nutritional health, including macro- (e.g., protein, fatty acids) and micronutrient (e.g., iron, zinc), to child-bearing age women as they enter pregnancy. Postnatally, the provision of human milk and maintenance of an optimal status of critical nutrients are key prevention strategies. As such, common gestational conditions such as maternal hypertension, diabetes, and obesity are major risk factors for early-life ID in a high-incomed society [3,88,89]. In particular, ID risk is double among Hispanic and Black child-bearing age women in the US, presenting a possible factor contributing to the multi-generational health disparity within these populations [41]. Thus, alternative treatment strategies may be successful. In this regard, dietary epigenetic modulators such as choline and folate are promising with a potentially beneficial therapeutic effect on neurodevelopment in the context of nutritional deficiency [18,90]; however, it is too soon to propose the use of these supplements as therapeutic interventions. Given the substantial changes in gene expression and the epigenome that can have a long-lasting adverse effect on fetal choline supplementation [18], more studies are needed to optimize the timing, dose, and duration of these agents.

## Figures and Tables

**Figure 1 nutrients-13-03857-f001:**
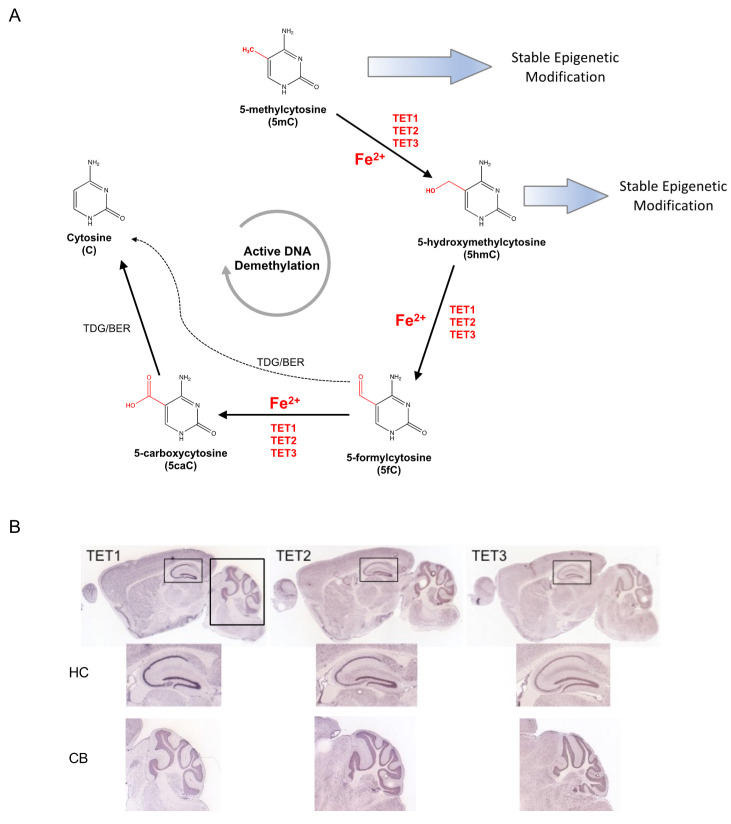
Iron-dependent Ten-Eleven Translocation (TET) proteins influence the epigenetic modifications of DNA methylation and hydroxymethylation. (**A**) Covalently modified cytosine bases are a major class of epigenetic modification. 5-methylcytosine (5mC) can be maintained long-term as a stable epigenetic modification or hydroxylated by the iron-dependent TET methylcytosine dioxygenases to 5-hydroxymethylcytosine (5hmC). 5hmC can serve as a stable epigenetic modification or an intermediate in the active DNA demethylation pathway, which reverses modified cytosines to unmodified states by serial modifications mediated in by TETs and thymine DNA glycosylase (TDG). (**B**) Expression of TET dioxygenases in the adult mouse brain. All three TET are highly expressed in the mouse hippocampus (HC) and cerebellum (CB). Images modified from Allen Mouse Brain Atlas [67].

**Figure 2 nutrients-13-03857-f002:**
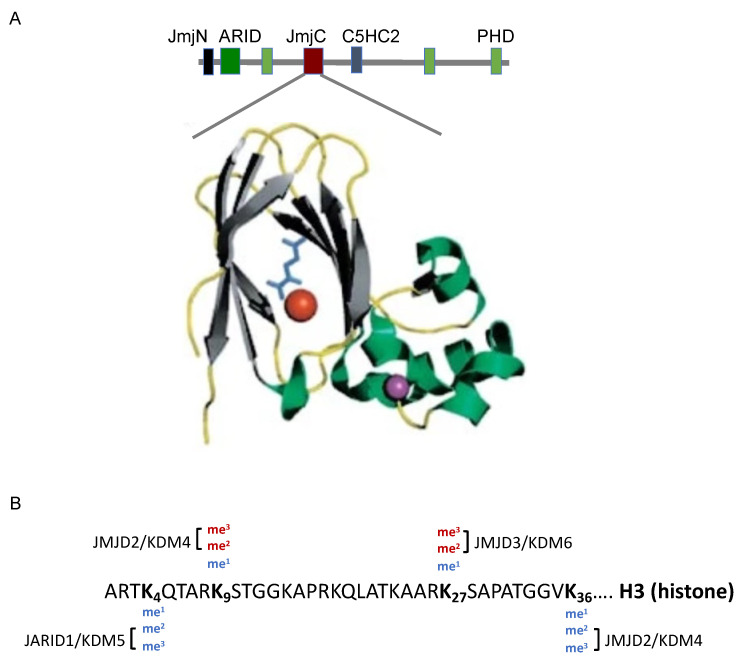
Iron-dependent jumonji and AT-rich interaction domain-containing (JARID) histone demethylase. (**A**) Illustration showing domains within JARID polypeptide. The Jumonji C (JmjC) domain is illustrated by 3D-modeling showing the iron (red) binding pocket with the co-factor alpha-ketoglutarate (blue). Image adapted from Klose, Kallin, and Zhang, Nature Reviews Genetics, 2006 [82]. (**B**) Major histone methylation sites within histone H3. Illustration depicting specific lysine (K) residues with histone demethylases (JARID/KDM). Enrichment of methylated K4 and K36 (blue) is associated with active transcription, whereas enrichment of methylated K9 and K27 (red) is associated with gene silencing (Adapted from Pedersen and Helin, 2010 [71]).

**Figure 3 nutrients-13-03857-f003:**
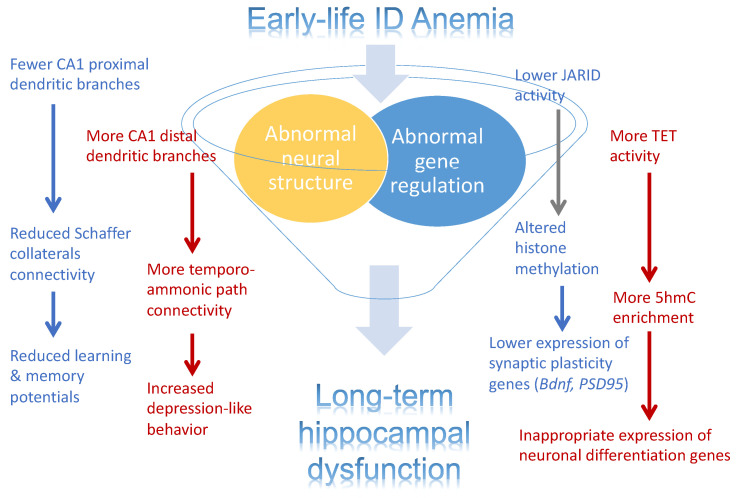
Effects of early-life iron deficiency (ID) anemia on the rat hippocampus. Illustration depicting the effects on early-life ID anemia on hippocampal structure and gene expression. Abnormal structure development, inappropriate gene expression, and their interaction are contributing factors to the abnormal hippocampal function in adult rats that were iron-deficient during fetal and neonatal life.

## Data Availability

Not applicable.

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
