# Peer review of "Early-Life Iron Deficiency Anemia Programs the Hippocampal Epigenomic Landscape"

_nutrients, 2021, doi:10.3390/nu13113857_

Round 1

Reviewer 1 Report

The manuscript "Early-Life Iron Deficiency Anemia Programs the Hippocampal Epigenomic Landscape" by Barks et al.  is a comprehensive review regarding the  significance of early-life iron deficiency anemia on neurodevelopment and risk for neuropsychiatric disorders. The review is well organized and discussed, and the references are appropriate and updated. Therefore, this review may be a good guide for future research on the role of the genes and methods to prevent fetal-neonatal iron deficiency, in order to minimize the risk of neurobehavioral deficit.

The suggestions to the authors are:

  1. Adjust the lenght of the abstract according to the instructions for authors (up to 200 words)
  2. Add the keywords.

Reviewer 2 Report

In the present review article Authors described very important problem concerning the effect of iron deficiency, in prenatal and early postnatal life, on molecular mechanism of nervous system development. The topic of the review is interesting and timely. Article is written well and comprehensible and also is prepared friendly for readers.

 I have only a few minor comments.

  1.     In the present article Authors focused on the effects of iron deficiency in the prenatal/early neonatal life. In my opinion it is worthy add a short chapter about placental iron transport in humans illustrated by appropriate scheme.

 2. In the chapter 2.3.2 (l. 222) Authors claimed that “Removal of methyl groups from lysine residues is catalyzed by the JARID protein family, which absolutely requires iron for their catalytic activity”, moreover Authors showed diagram of the tertiary structure of the JARID protein (Fig 2). Of course I agree with this opinion, but please say, in a few sentences, a little more about the iron (Fe2+) function in stabilisation of JARID protein structure (Chen et al. 2006).   

 3.     Please also add  Keywords 
